# Investigation on the Fiber Orientation Distributions and Their Influence on the Mechanical Property of the Co-Injection Molding Products

**DOI:** 10.3390/polym12010024

**Published:** 2019-12-20

**Authors:** Chao-Tsai Huang, Xuan-Wei Chen, Wei-Wen Fu

**Affiliations:** Department of Chemical and Materials Engineering, Tamkang University, No. 151, Yingzhuan Rd., Tamsui Dist., New Taipei City 25137, Taiwan; willych1014@gmail.com (X.-W.C.); fu840501@gmail.com (W.-W.F.)

**Keywords:** co-injection molding, fiber reinforced plastics (FRP), fiber orientation distribution (FOD), micro-computerized tomography (μ-CT) scan technology

## Abstract

In recent years, due to the rapid development of industrial lightweight technology, composite materials based on fiber reinforced plastics (FRP) have been widely used in the industry. However, the environmental impact of the FRPs is higher each year. To overcome this impact, co-injection molding could be one of the good solutions. But how to make the suitable control on the skin/core ratio and how to manage the glass fiber orientation features are still significant challenges. In this study, we have applied both computer-aided engineering (CAE) simulation and experimental methods to investigate the fiber feature in a co-injection system. Specifically, the fiber orientation distributions and their influence on the tensile properties for the single-shot and co-injection molding have been discovered. Results show that based on the 60:40 of skin/core ratio and same materials, the tensile properties of the co-injection system, including tensile stress and modulus, are a little weaker than that of the single-shot system. This is due to the overall fiber orientation tensor at flow direction (A_11_) of the co-injection system being lower than that of the single-shot system. Moreover, to discover and verify the influence of the fiber orientation features, the fiber orientation distributions (FOD) of both the co-injection and single-shot systems have been observed using micro-computerized tomography (μ-CT) technology to scan the internal structures. The scanned images were further utilizing Avizo software to perform image analyses to rebuild the fiber structure. Specifically, the fiber orientation tensor at flow direction (A_11_) of the co-injection system is about 89% of that of the single-shot system in the testing conditions. This is because the co-injection part has lower tensile properties. Furthermore, the difference of the fiber orientation tensor at flow direction (A_11_) between the co-injection and the single-shot systems is further verified based on the fiber morphology of the μ-CT scanned image. The observed result is consistent with that of the FOD estimation using μ-CT scan plus image analysis.

## 1. Introduction

Due to its excellent properties, fiber-reinforced plastics (FRP) material has been applied in industry for years, especially as one of the major lightweight technologies for automotive or aerospace products [1,2]. Specifically, the market of composite products is expected to achieve an estimated $40.2 billion by 2024. The compound annual rate (CAGR) from 2019 to 2024 is about 3.3% [3,4]. Moreover, according to the Composite Industry Market report in 2018, around 1.141 million tons of glass-FRP composite materials were produced in Europe [5]. In the United States, the demand for FRP was 4.3 billion pounds in 2017 [3]. To handle the recycling of FRP composites, there are three primary methods, including mechanical, thermal, and chemical processes [3,6]. However, since the presence of the fiber makes the microstructures inside FRP more complicated than that of general thermoplastics, it causes the recycling of FRP to be very difficult. As higher amounts of FRP are consumed, higher environmental impacts are needed to be addressed now and in future [7]. To overcome this impact, co-injection molding, based on the mechanical recycling process, could be a good solution.

Co-injection molding is commonly used as a daily accessory, and in many other contexts. Basically, the co-injection technology can provide several advantages with integrating materials to reduce cost. This technology also allows for the reuse of materials, offers an upgrade in production efficiency, and can make raw skin/recycled core structures. However, there are some challenges for co-injection processes. For example, how to visualize and control suitable skin/core material distribution is very difficult during the processing. To deal with this complicated process, there is some literature that can be used as guidelines. Seldén [8] monitored different process conditions and found out the skin-to-core material ratio is the main factor causing break-through problems. Moreover, the correlation between internal material distributions, process condition, and material property are discussed in many previous studies [9,10,11]. The cavity-filling ratio of skin material determines the break-through location. Material viscosity and filling rate affect uniformity of core material distribution. Furthermore, since the geometrical structure of real products is very complicated, the progress of the co-injection processing becomes much more of a challenge. To simulate this situation, Yang and Yokoi [12] proposed a co-injection with a multi-cavity molding system with a fork structure. They found that the core flow pattern in the fork structure is strongly affected by injection flow rate. It is also affected by material property. However, their work is not comprehensive yet because they didn’t discuss the flow behavior at the end of filling for the multi-cavity system. Later, Huang [13] and Huang et al. [14] verified that in multi-cavity co-injection systems, the skin-to-core material ratio is still the main factor to dominant the break-through phenomena, while the injection flow rate can be used to adjust the core penetration uniformity. Indeed, the core penetration prediction in the real co-injection system is still very challenging.

On the other hand, as mentioned earlier, due to its excellent properties, the fiber-reinforced plastic (FRP) material has been applied in many industries. The reason that FRP has so much potential is because of the functions of fibers’ microstructures, including their orientation, length, and concentration. It is expected that those microstructures will further affect the final shrinkage and warpage of the single-shot and co-injected parts. Moreover, how to manage the glass fiber orientation features and their influence in the recycling of the FRP products are still very complicated. To realize how fiber structures would influence the mechanical properties of the finished parts, Thomason et al. [15,16] studied the phenomenon experimentally and proposed some empirical standards of fiber length to guarantee enough impact strength for automobiles. Moreover, some researchers have studied the important fiber microstructure variables, including fiber orientation, fiber length, and fiber concentration, to determine the level of effectiveness of mechanical property enhancement [17,18,19]. Cilleruelo et al. further considered the effect of carbon black and nucleating agents on impact properties experimentally [20]. Indeed, the microstructures of fibers were still very difficult to validate in this work. To understand why the fiber orientation is changed, Folgar and Tucker [21], Advani and Tucker [22], and Advani [23] proposed numerical models to predict fiber orientation in short fiber. Using their short fiber model, the fiber orientation distribution (FOD) could be predicted reasonably. As the demand on the impact property increases, the longer fiber length retained in the final injected part is in greater demand. In recent years, some researchers have extended the numerical prediction capability and experimental validation to the long fiber orientation [24,25,26,27]. However, no matter how short or long the fibers, to the best of our knowledge, very few researchers have discovered the connection between the fiber microstructures to the mechanical properties of the final injection parts quantitatively. Moreover, knowing how to catch these fiber microstructures in the final parts is still not easy in reality. In general, there are two methods which have been utilized. One is optical section method, and the other is micro-computerized tomography (μ-CT) scan method [28]. Some researchers [28,29,30] have applied the micro-computerized tomography (μ-CT) scan method to get through finished parts. The μ-CT is a relatively new method to measure the fiber orientation distribution in FRP parts. It is a non-destructive testing method, but the procedure was still not easy to handle. This is especially because there are a huge number of images created after using μ-CT technology, and further image analysis is another key issue to deal with. Overall, this is to say; the ways in which fiber orientation provides the reinforced features in co-injection molding is still not fully understood in reality. In addition, the exact working function of the fibers is not easily visualized and managed. 

In this study, we have applied both CAE simulation and experimental methods to investigate the interface (between skin and core) penetration behavior. Furthermore, the connection between the inside FOD variation and the tensile properties of the single-shot and co-injection molding based on the standard tensile bar (ASTM D638 TYPE V) system has been studied. To give better understanding, the main content of this paper is organized as follows. The theoretical background is presented in Section 2. Section 3 describes the model and related information. It will discuss the simulation model and experimental equipment separately. Then, the results and discussion are in Section 4. Finally, the brief conclusion will be addressed in Section 5.

## 2. Theoretical Background and Numerical Method

### 2.1. Model for Co-Injection Molding

The numerical simulation was conducted using the Moldex3D R16^®^ software. Both the skin and core materials are considered to be compressible, generalized Newtonian fluid. Surface tension at the melt front is neglected. The governing equations for 3D transient non-isothermal motion are [13]:(1)$$\frac{\partial \rho }{\partial t}+\nabla \cdot \rho u=0$$
(2)$$\frac{\partial }{\partial t}\left(\rho u\right)+\nabla \cdot \left(\rho uu+\tau \right)=-\nabla p+\rho g$$
(3)$$\rho C_{P}\left(\frac{\partial T}{\partial t}+u\cdot \nabla T\right)=\nabla \cdot \left(k\nabla T\right)+\eta {\dot{\gamma }}^{2}$$
where ρ is density; **u** is velocity vector; *t* is time; τ is total stress tensor; **u** is acceleration vector of gravity; *p* is pressure; η is viscosity; *C_p_* is specific heat; *T* is temperature; *k* is thermal conductivity; $\dot{\gamma }$ is shear rate. For the polymer melt, the stress tensor can be expressed as:(4)$$\tau =-\eta \left(\nabla u+\nabla u^{T}\right)$$

The modified-Cross model with Arrhenius temperature dependence is employed to describe the viscosity of polymer melt:(5)$$\eta \left(T,\dot{\gamma }\right)=\frac{\eta_{o}\left(T\right)}{1+{\left(\eta_{o}\dot{\gamma }/\tau^{*}\right)}^{1-n}}$$
with
(6)$$\eta_{o}\left(T\right)=BExp\left(\frac{T_{b}}{T}\right)$$
where *n* is the power law index, $\eta_{o}$ is the zero shear viscosity, $\tau^{*}$ is the parameter that describes the transition region between zero shear rate, and the power law region of the viscosity curve. 

A volume fraction function, *f*i, is introduced to specify the evolution of the polymer/air front (i = 1) and skin/core front (i = 2) interfaces. Here, *f*i = 0 is defined as the no-filled region, *f* = 1 as the fully-filled region, and finally the interfacial front is located within cells of an *f* value between 0 and 1. The advancement of *f* over time is governed by the following transport equation:(7)$$\frac{\partial f_{i}}{\partial t}+\nabla \cdot \left(uf_{i}\right)=0$$

During the polymer melt filling phase, the velocity and temperature are specified at the mold inlet. While the core material is injected, the flow rate setting is specified at the mold inlet. On the mold wall, the non-slip boundary condition is applied, and fixed mold wall temperature is assumed. 

### 2.2. Model for Fiber Orientation Distribution 

The fiber orientation is described as follows. A single fiber is regarded as an axisymmetric bond with rigidness. The bond’s orientation unit vector **p** along its axis direction can described as the fiber orientation. Orientation state of a group of fibers is given by second moment tensor,
(8)A = ∮ψ(p)pp dp
where ψ(p) is the probability density distribution function over orientation space; ***p*** is the definition of the orientation vector, as shown in Figure 1. Tensor **A**_4_ is a fourth order orientation tensor, defined as:(9)$$A_{4}\;=\;\oint \psi \left(p\right)pppp\;dp$$
where this tensor is also symmetric. The acceptable calculation is obtained through the eigenvalue-based optimal fitting approximation of the orthotropic closure family. To handle this complicated tensor system, Tseng et al. [25,26] developed a new fiber orientation model to couple with Jeffery’s hydrodynamic (HD) model, namely, the iARD-RPR model (known as Improved Anisotropic Rotary Diffusion model combined with Retarding Principal Rate model),
(10)$$\dot{A}\;=\;{\dot{A}}_{\mathrm{HD}}\;+\;{\dot{A}}_{\mathrm{iARD}}\left(C_{I},\;C_{M}\right)\;+\;{\dot{A}}_{\mathrm{RPR}}\left(\alpha \right)$$
where $\dot{A}$ represents the material derivative of A. Parameters C_I_ and C_M_ describe the fiber–fiber interaction and fiber–matrix interaction, while parameter α can slow down a response of fiber orientation. Details of the RPR model and the iARD model are available elsewhere [21,22].
(11)$${\dot{A}}_{\mathrm{HD}}\;=\;\left(W\cdot A\;-\;A\cdot W\right)\;+\;\xi \left(D\cdot A\;+\;A\cdot D\;-\;2A_{4}:D\right)$$
where **W** and **D** are the vorticity tensor and rate-of-deformation tensor, respectively. ξ is a shape factor of a particle.

## 3. Geometrical Model and Related Information

### 3.1. Simulation Model and Related Information

The geometry model and dimensions of the runner and cavity are shown as in Figure 2. Specifically, it is based on ASTM D638 Type V standard specimen with dimension of 63.5 mm × 9.53 mm × 5.3 mm (see Figure 2a). The moldbase and cooling channel layout is exhibited in Figure 2b. The meshed model is shown in Figure 2c. The associated mesh type and element count are listed in Table 1. In addition, the numerical convergence testing regarding the mesh resolution is shown in Figure 3. Clearly, when it is kept with 20-layer or higher in thickness direction, the sprue pressure history curve is almost unchanged. Hence, in this study, the major mesh type is hexahedron. The selected mesh model is Mesh 4 which has 20-layer in thickness direction and 252,720 total element counts. Moreover, the process conditions for the co-injection process and the counterpart single-shot injection are listed in Table 2 and Table 3. Briefly, the filling time is 0.3 s; melt temperature is 230 °C; mold temperature 35 °C; skin-to-core switch over is at 60% by volume. The material used is PP Globalene SF7351 which has 30% short fiber content. Moreover, in order to conduct the interface (between skin and core) penetration behavior and also study the fiber orientation variation dynamically, some measuring nodes have been specified, as shown in Figure 4. Specifically, there are three measuring nodes named A, B, and C. During the co-injection process, when the interface between skin and core arrives at point A, it is specified as time t1 (see Figure 4b). Similarly, when the interface arrives at point B, and then point C, it is specified as t2 and t3, respectively, as displayed in Figure 4c,d.

### 3.2. Experimental Model and Related Information

In order to realize what happens during the co-injection molding physically, the real co-injection system and the mold were constructed, as shown in Figure 5. The machine model is TA-4.0ST-2ST-80T made by Ta Ai Machinery Co. Ltd. from Taiwan (Figure 5a). The cavity with the same dimension as that of the simulation is listed in Figure 5b. Moreover, to observe the real fiber orientation behavior, micro-computerized tomography (μ-CT) technology was performed using Bruker Skyscan 2211 with 100 kV, 8.3 W, and a resolution of 7 μm, supported by MCL Multiscale X-ray CT laboratory, Industrial Technology Research Institute, Taiwan. Moreover, to realize how the mechanical properties have been affected due to the different processing, the tensile test was performed. The universal tensile testing machine of LS1 high precision testing machine model supplied by Lloyd company was used as shown in Figure 6. The testing procedures are based on [31]. The definition of the dimension parameter and the specific amount are listed in Figure 7 and the Table 4. During each testing, at each time period, the deformation and the associated force were recorded. Then, the deformation and associated force were transferred into stress and strain, based on Equations (12) and (13), described below. The stress and strain obtained in Equations (12) and (13) can be used to make the stress–strain curve. The tensile stress (σ) is obtained by dividing the force (*F*) by the initial cross-section area of the narrow portion (*A*).
(12)$$\sigma \;=\frac{F}{A}=\;\frac{F}{\left(W_{1}\;\times \;H\right)}$$

The elongation (ε) is obtained by dividing the deformation (ΔL) by the length of narrow portion and stated as a percentage.
(13)$$\varepsilon \;=\frac{\Delta L}{L_{1}}\;\times \;100\%$$

The tensile modulus (E) is calculated from the secant between 0.05% and 0.25% strain of the averaged stress–strain curve.
(14)$$E\;=\frac{\sigma_{0.25\%\;}-\;\sigma_{0.05\%}}{\varepsilon_{0.25\%}\;-\;\varepsilon_{0.05\%}}$$

## 4. Results and Discussion

### 4.1. Skin/Core Ratio Effect 

Figure 8 shows the numerical simulation of the movement of the core interface (between skin and core) at various skin/core ratio settings. When the ratio is from 90:10 to 60:40, the higher the ratio, the longer core penetration could be obtained. However, when the ratio is 50:50, the skin break-through phenomenon happened close to the end of the cavity. Furthermore, when the skin/core ratio is increased from 50:50 to 10:90, the skin break-through area was extended. Alongside this, the final core penetration location was be moved from the end of the cavity to the beginning of the cavity, as far as to the runner. This clearly shows how the break-through location is very sensitive to the skin/core ratio in co-injection molding. In addition, in some applications (for example, in plastic recycling), it is expected to that there is more core material covering the inside of the co-injection product to prevent break-through happening. However, to avoid the break-through phenomena happen, a skin/core ratio of 60:40 will be a good choice for our future studies.

### 4.2. Fiber Orientation Distribution (FOD) Prediction 

Moreover, to study the fiber effect in FRP products, the fiber orientation distribution (FOD) inside the cavity during single-shot molding has been investigated. For comparison purposes, as the core interface touched point A (time t1), we selected the same melt front time for both single-shot and co-injection systems, as shown in Figure 9a. Meanwhile, Figure 9b presents the fiber orientation tensor components A_11_, A_22_, and A_33_ for the single-shot molded specimen. Specifically, from the top surface through the frozen layer to the core layer at the central line of the specimen, the fiber orientation tensor at flow direction (A_11_) will be slightly increased from 0.75 to 0.85, and then decreased to 0.25. At the same time, the fiber orientation tensor in the cross-flow direction (A_22_) will be slightly decreased from 0.20 to 0.15, and then increased to 0.5. Furthermore, Figure 9c shows the fiber orientation tensor (A_11_) at point A through different time periods (from t1 to t3). In Figure 9c, the fiber orientation tensor (A_11_) is almost the same from t1 to t3 in the single-shot system. It can be inferred that the fiber orientation is strongly affected by the frozen layer and the adjacent shear layer during the conventional injection molding. As long as the frozen layer forms, it influences the shear layer to build a strong A_11_ fiber orientation tensor. Meanwhile, since the FOD calculation in co-injection system is not developed successfully yet, that FOD of the co-injection system will be discovered by using μ-CT plus image analysis technology. The details will be addressed in a later section.

### 4.3. Experimental Investigation and Validation 

#### 4.3.1. Short Shot Validation

Figure 10 presents the short shot testing at the skin/core ratio of 60:40 for both the simulation and the experiment. Based on the careful evaluation from 46% to 100% of filling, it is clear that the flow front and interface penetration for the simulation prediction is quite closely matched with the experimental observation. 

#### 4.3.2. Break-Through Study and Validation

Moreover, Figure 11 shows the experimental validation of the skin/core ratio effect, specifically for the evaluation of the break-through locations. Basically, from a skin/core ratio of 90:10 to 60:40, no break-through phenomena happened. However, when the skin/core ratio is from 50:50 to 10:90; the more core material injected, the larger the penetration area that is observed. Figure 11b shows the break-through behavior for the skin/core ratio of 40:60. The results show that both simulation and experiment are in close agreement. In addition, the final core penetration location will be moved from the end of the cavity to the beginning of the cavity, even to the runner, when the core material ratio is increased. Overall, this is to say that the observation is matched with the simulation prediction very well.

#### 4.3.3. Fiber Morphology Observation

In order to understand the fiber morphologies and their difference between the single-shot and co-injection systems, the μ-CT scanned images (sliced plane) at different thickness locations have been selected. Figure 12 defines the locations of the selected planes to observe the fiber morphology. Three thickness locations from the top surface have been selected, that is, thickness th = 0.5 mm, 1.0 mm, and 1.75 mm, respectively. Moreover, the fiber morphology inside the co-injected parts can be observed, as shown in Figure 13. Specifically, Figure 13(a1) presents the numerical prediction of the skin/core interface. Since there is no core material observed at th = 0.5 mm, there is no skin/core interface. Figure 13(b1) shows the fiber morphology at thickness th = 0.5 mm experimentally. At this plane, no core material existed. This shows that the fibers are strongly aligned in flow direction near the frozen-layer and shear-layer, but also presents the cross-flow direction in the center core region. Furthermore, when it is moved to thickness th = 1.0 mm, as shown in Figure 13(a2,b2), the skin/core structure appears. In order to trace the skin/core interface and detect the differences across the interface boundary in co-injection molded parts, we have applied numerical simulation (in Figure 13(a2)) to specify the interface first. Then, the marked boundary line can be inserted into the sliced plane of the real scanned image as in Figure 13(b2). When we focused on the areas across the interface boundary from the outer material (skin) to the inner material (core), there was no significant difference across the interface boundary. Similarly, we even moved to the central thickness portion (th = 1.75 mm), as shown in Figure 13(a3,b3). There was no significant difference across the interface boundary from the skin to core materials. Hence, when an FRP co-injection is performed using two of the same materials, the skin/core interface is not significant. 

Moreover, it is worth discovering what the difference was in fiber morphology between the single-shot and co-injection FRPs parts. Figure 14 shows the observation on the fiber morphology for the single-shot and co-injected parts at different thickness locations. Specifically, one region with (7 mm × 3 mm) has been selected for comparison. In Figure 14(a1,b1), from frozen-layer to central core-layer, no core material existed at th = 0.5 mm. In this region, the strongly aligned fiber area of the single-shot is larger than that of the co-injection system. Similarly, when we focused on the fiber morphology at th = 1.0 mm, as shown in Figure 14(a2,b2), the fibers presented similar behavior, as shown in Figure 14(a1,b1). Finally, the similar trend can be observed in Figure 14(a3,b3). Hence, when it generates one boundary interface via skin/core sequential co-injection molding using two of the same FRPs, the flow field drives the fibers to orientate less in the flow direction. Even the change is small, and this behavior will further affect the mechanical property of the final parts.

#### 4.3.4. Fiber Orientation Distribution (FOD) Estimation and Validation

Furthermore, it is worth connecting fiber reinforced features to the product quality. The fiber microstructures can be characterized partially based on fiber orientation distribution (FOD). Figure 15a shows the simulation prediction of the FOD at end of filling (t = t3). From the top surface through the frozen layer to the core layer, at central line of the specimen, the fiber orientation tensor at flow direction (A_1**1**_) will be increased slightly from 0.75 to 0.85, and then decreased to 0.25. At the same time, the fiber orientation tensor at cross direction (A_22_) will be decreased slightly from 0.20 to 0.15, and then increased to 0.5. This result is consistent with that observed in Figure 14(a1,a3). Meanwhile, the real specimens are scanned by μ-CT to get the detailed inner microstructure images. Then, those images are further analyzed using AVIZO software to rebuild the structures. Figure 15b presents the FOD of the real measurement, from top surface through frozen layer to core layer at central line of the specimen. The fiber orientation tensor at flow direction (A_11_) is increased slightly from 0.7 to 0.78, and then decreased to 0.2. At the same time, the fiber orientation tensor at cross direction (A_22_) will slightly be decreased from 0.30 to 0.2, and then increased to 0.7. Moreover, there is a little difference between the numerical prediction and experimental study from normalized thickness 0 to 0.2 region. This could be due to the shrinkage of the co-injection parts which will further affect the direction of the FOD along the frozen layer. Overall, the FOD via the real measurement is in a reasonable agreement with that of simulation prediction.

Moreover, the detailed FOD behavior at flow direction (A_11_) and cross-flow direction (A_22_) for both single-shot and co-injection were investigated. Figure 16a shows that the flow direction FOD tensor (A_11_) of the co-injection is lower than that of the single-shot at the central portion. Furthermore, the overall fiber orientation capability can be described by the integration of the area under the FOD curve. For example, the area below the curve of the co-injection to that of the single-shot is (0.47/0.53 = 0.89). That is to say, the flow direction FOD tensor (A_11_) of the co-injection is about 89% of that of the single-shot. On the other hand, in Figure 16b, the cross-flow direction FOD tensor (A_22_) of the co-injection is higher than that of the single-shot at the central portion. Specifically, the area below the curve of the co-injection molding to that of the single-shot is (0.44/0.38 = 1.16). The cross-flow direction FOD tensor (A_22_) of the co-injection is about 116% of that of the single-shot. From the above results the fiber orientation features of the co-injection molding can be quantified as “the flow direction A_11_ is decreased by 11%, and cross-flow direction A_22_ is increased by 16%.” It is inferred that the FOD variation of the co-injection system is due to the presence of the skin layer which can reduce the influence of the solid boundary for changing the fiber alignment. 

#### 4.3.5. Tensile Property Measurement

Moreover, to understand the difference of the fiber reinforced performance between two processes, the mechanical properties based on standard tensile test can be applied. As mentioned earlier, the tensile testing was performed following the standard procedures described in Ref. [31]. Specifically, the specimen is installed into the machine holder (see Figure 6b) under a constant strain at 20 mm/min without pre-tensioning. For each system (single-shot or co-injection), five specimens have been used for the same testing. After finished five tests for each type, the average stress–strain behavior is presented in Figure 16. In Figure 17a, the stress–strain behavior of the single-shot injected part is a little higher than that of co-injection part. Moreover, the associated tensile properties including elongation at break, tensile strength, and tensile modulus are further discussed. Figure 17b shows that the single-shot sample has a larger elongation at break feature than that of the co-injection system, but it is not significant. Furthermore, the detailed tensile stresses and modulus are recorded in Figure 17c and Table 5. Obviously, the average tensile strength of the single-shot is a little higher than that of the co-injection molding by 1.6% (i.e., (86.07 − 84.69)/86.07 × 100% = 1.6%). As per our previous observation and conduction, overall, the fiber orientation tensor at flow direction (A_11_) of the single-shot is higher than that of the co-injection. Hence, the lower tensile strength and stress modulus of co-injection are consistent with that observation of fiber orientation variation due to the sequential co-injection process. However, the reduction of the tensile properties by co-injection molding is not significant. Hence, using co-injection molding to execute FRP recycling is feasible theoretically. Moreover, some connection between the fiber orientation and the mechanical properties of the final co-injection parts can be obtained.

## 5. Conclusions

In this study, we have applied both CAE simulation and an experimental study to investigate the fiber orientation distributions and their influence on the tensile properties for single-shot and co-injection molding. Specifically, we have obtained some connection between the fiber orientation and the mechanical properties of the final co-injection parts. Some key findings are as follows: The skin/core ratio of 60:40 can provide suitable core-layer penetration without break-through, numerically and experimentally. To discover and verify the influence of the fiber orientation features, the fiber orientation distributions (FOD) of both co-injection and single-shot systems have been observed using μ-CT technology to scan the internal structures, and then software used to perform image analyses for those scanned images. Specifically, the fiber orientation tensor at flow direction (A_11_) of the co-injection is about 89% of that of the single-shot in the testing conditions. The lower the A_11_, the lower the tensile property that is expected. The difference of the fiber orientation tensor at flow direction (A_11_) between the co-injection and the single-shot systems is further verified based on the fiber morphology of the μ-CT scanned image. The observed result is consistent with that of the FOD estimation using the μ-CT scan plus image analysis.To validate the FOD effect on the mechanical properties due to the co-injection, tensile testing was performed. The tensile strength and tensile modulus of the co-injection part is a little weaker than that of the single-shot system. The reason inferred is that the overall fiber orientation tensor at flow direction (A_11_) of the co-injection system is lower than that of the single-shot system. 

## Figures and Tables

**Figure 1 polymers-12-00024-f001:**
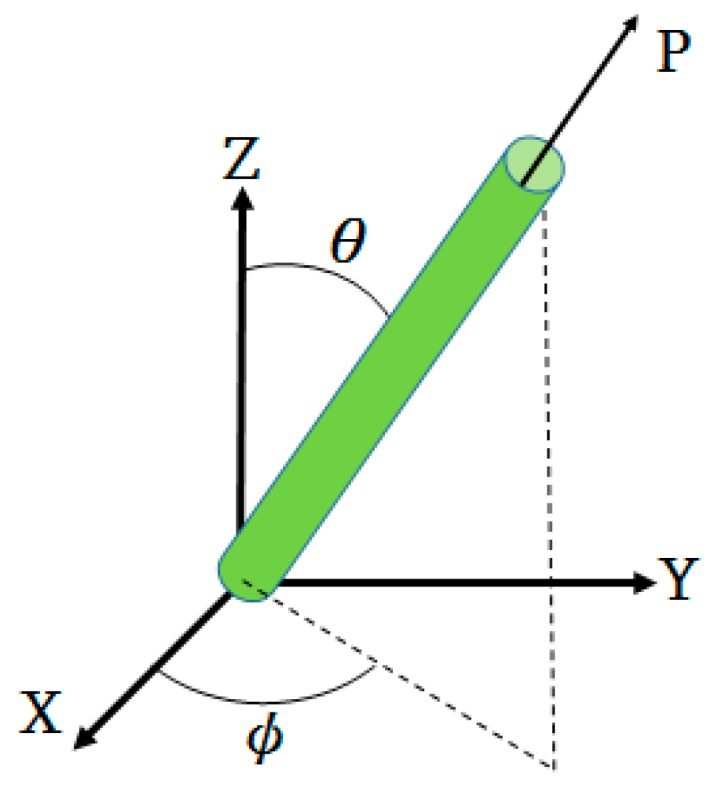
Definition of the orientation vector **p.**

**Figure 2 polymers-12-00024-f002:**
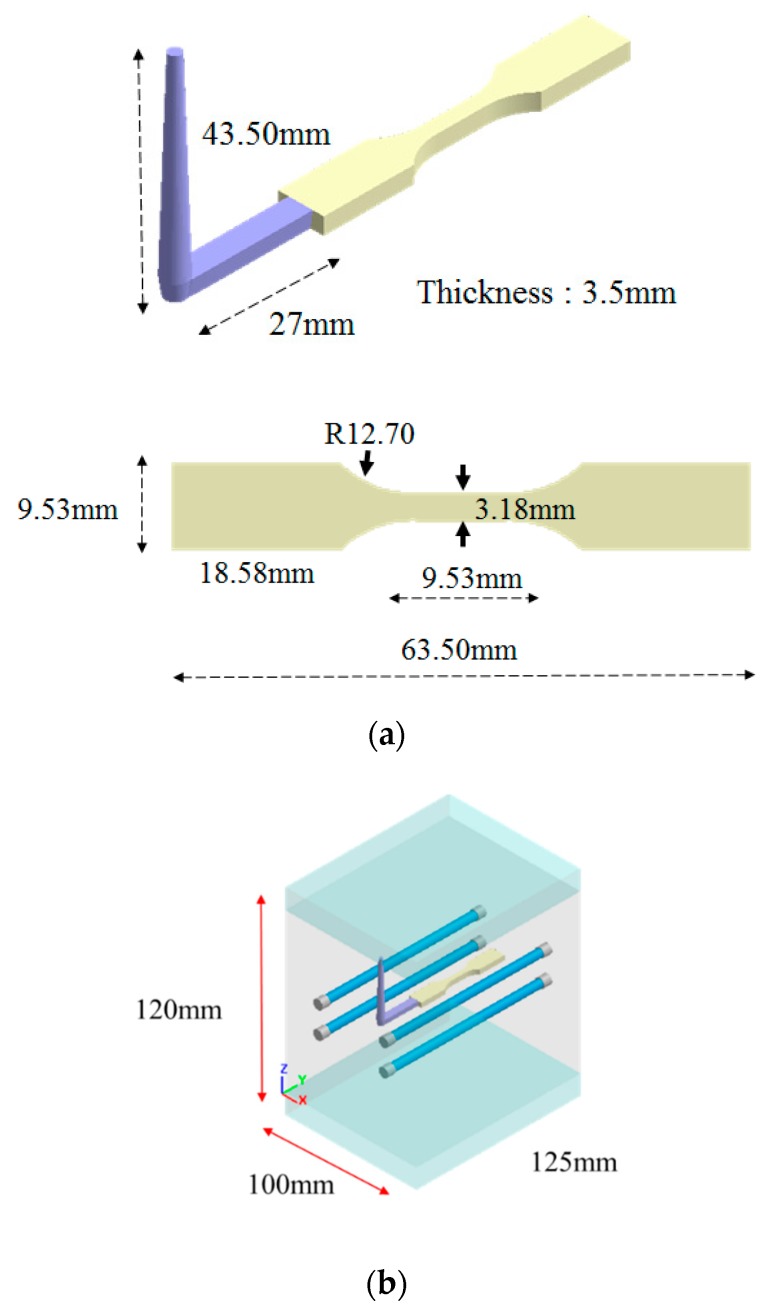
(**a**) Geometry model and dimensions, (**b**) moldbase and cooling channel layout, (**c**) the meshed model.

**Figure 3 polymers-12-00024-f003:**
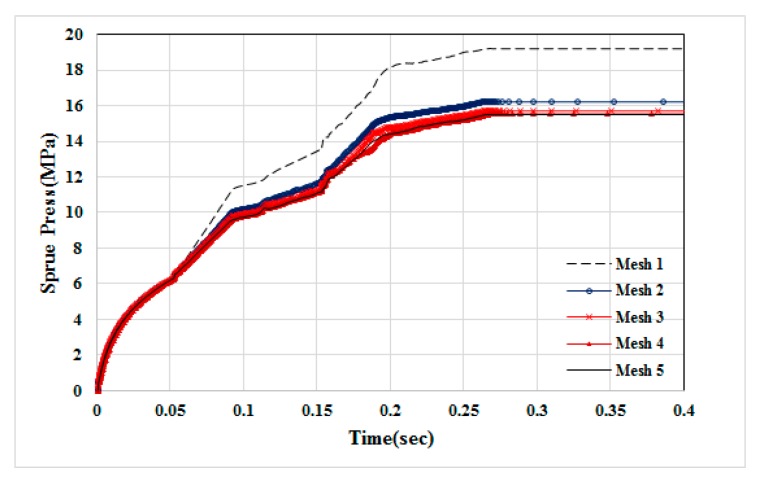
The numerical convergence testing for mesh type and resolution.

**Figure 4 polymers-12-00024-f004:**
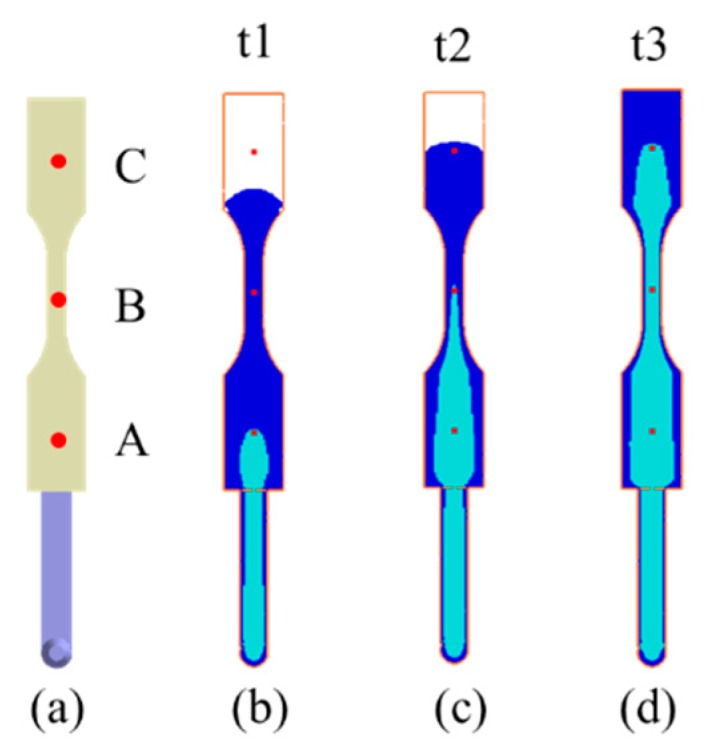
The locations of measuring modes: (**a**) Locations of three measuring nodes, (**b**) t1: The core interface arrives at point A, (**c**) t2: The core interface arrives at point B, (**d**) t3: The core interface arrives at point C.

**Figure 5 polymers-12-00024-f005:**
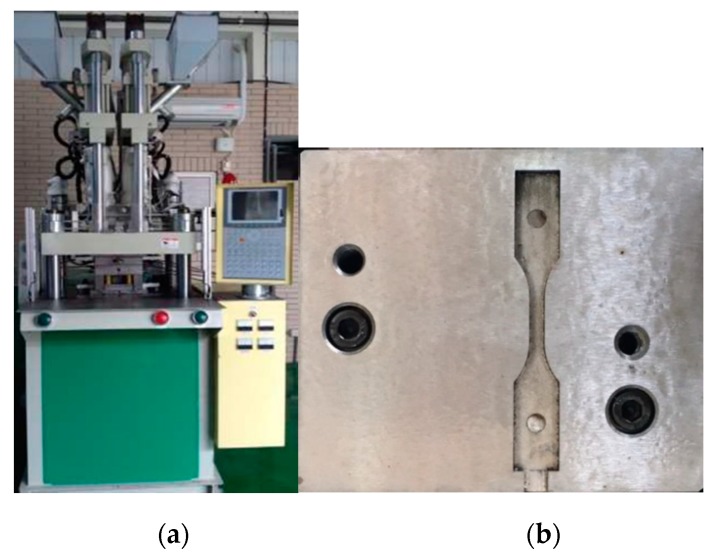
(**a**) Co-injection molding system, (**b**) the cavity.

**Figure 6 polymers-12-00024-f006:**
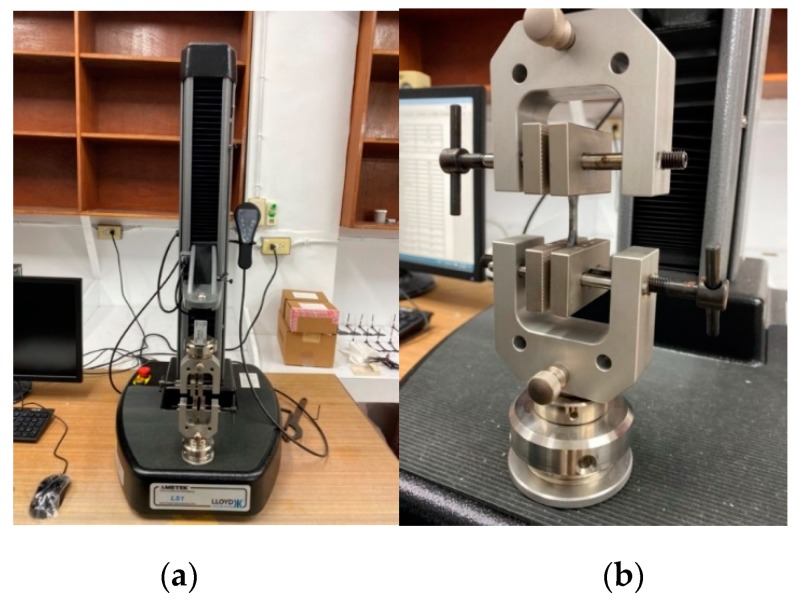
(**a**) The universal tensile testing machine, (**b**) the sample holder.

**Figure 7 polymers-12-00024-f007:**
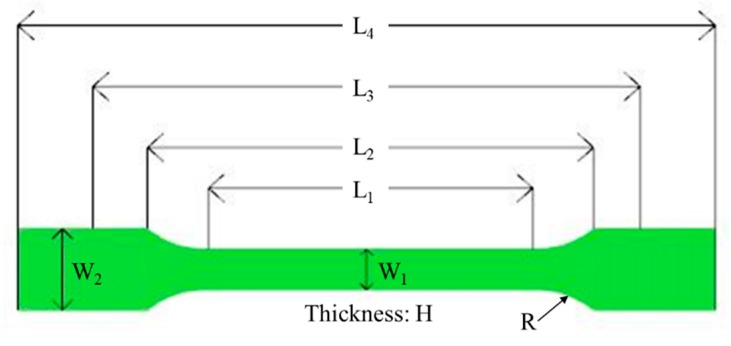
The parameter definition for the tensile test.

**Figure 8 polymers-12-00024-f008:**
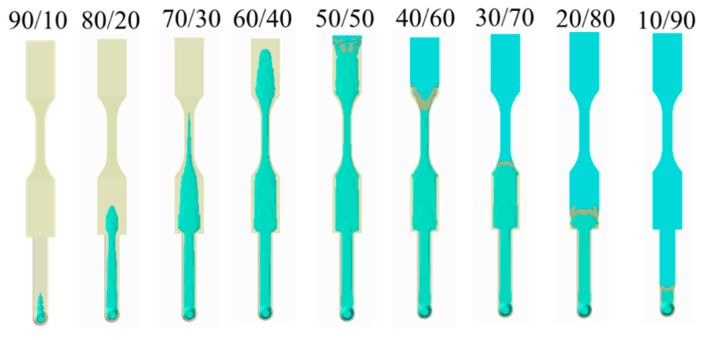
Core penetration behavior at various skin/core ratios, from 90:10 to 10:90.

**Figure 9 polymers-12-00024-f009:**
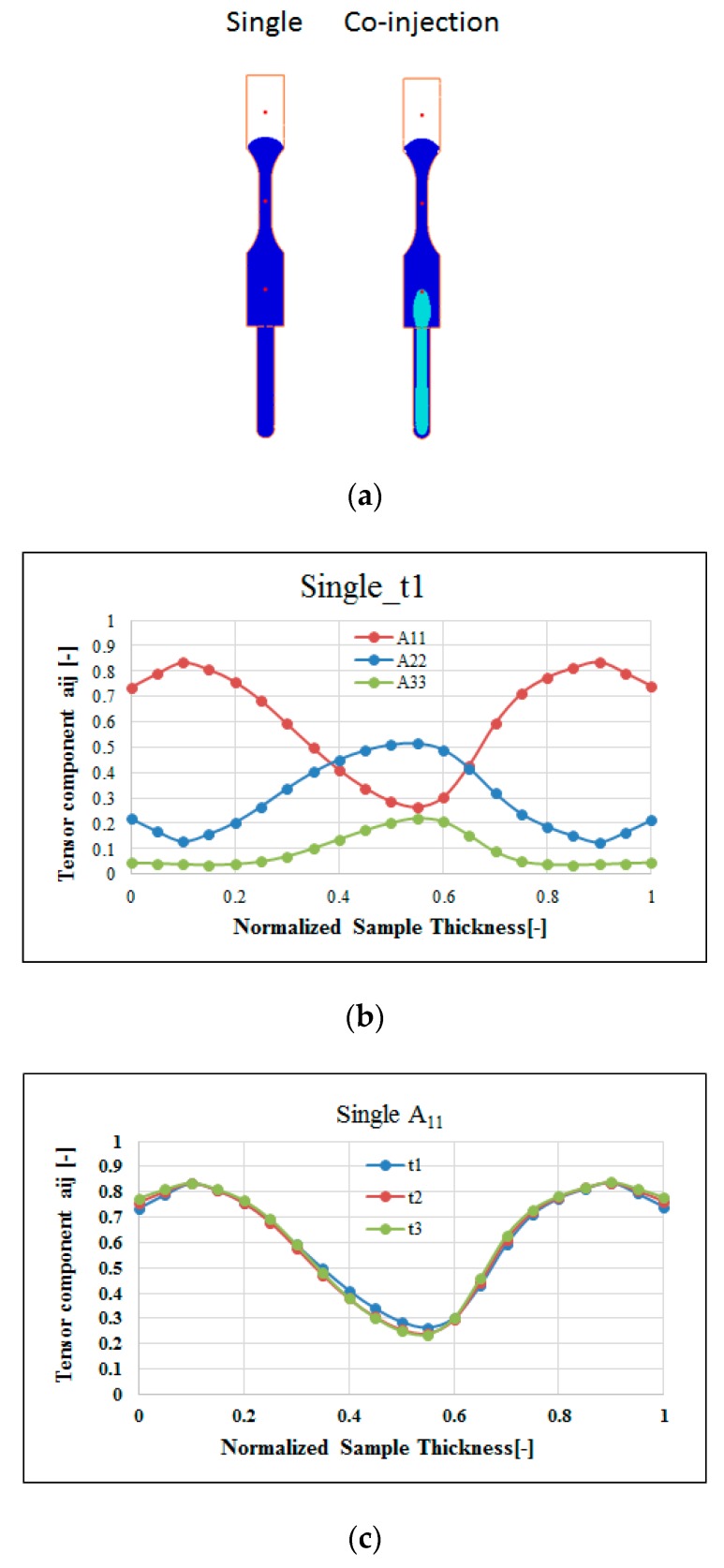
Fiber orientation features of the single-shot: (**a**) The time period as the flow front is the same for both single and co-injection, (**b**) the FOD for a single shot at point A at time t1, (**c**) the flow direction orientation tensor A_11_ at point A with different time period.

**Figure 10 polymers-12-00024-f010:**
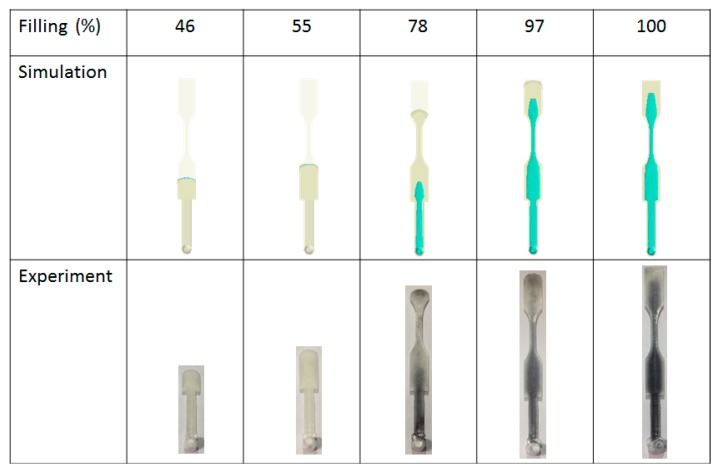
Short shot testing for simulation prediction and experimental study (at skin/core ratio = 40:60).

**Figure 11 polymers-12-00024-f011:**
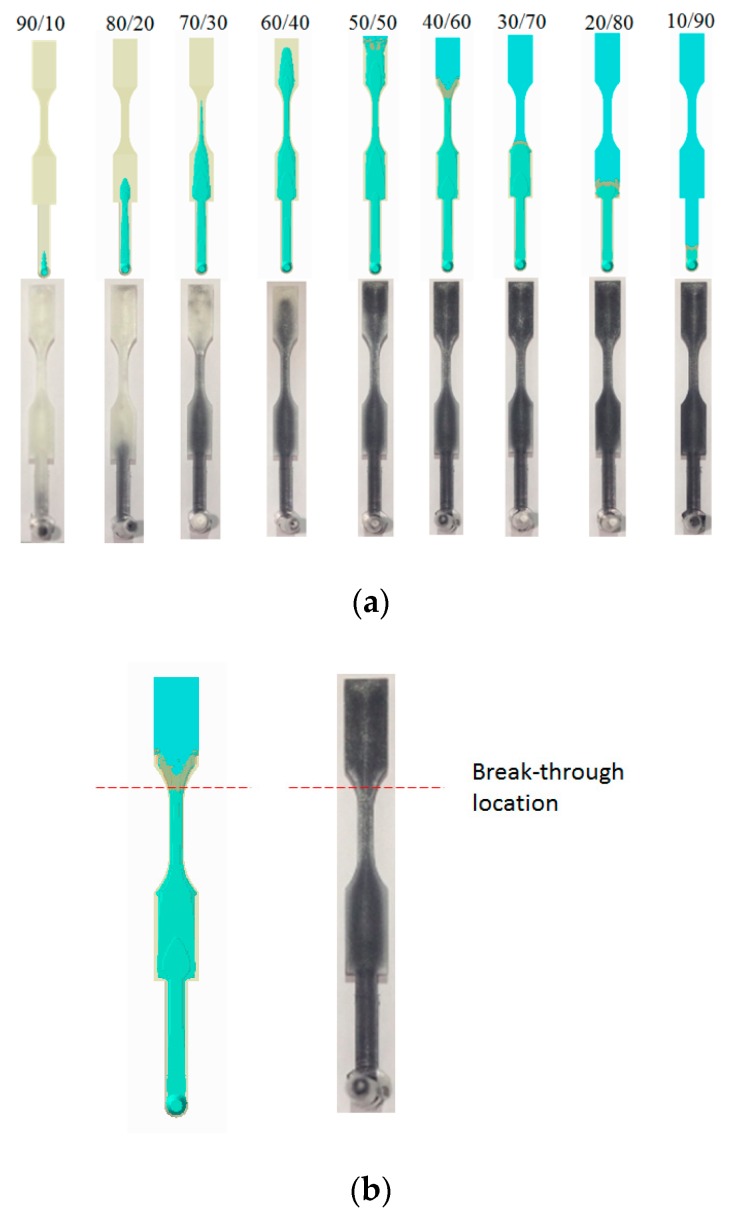
Experimental validation for skin/core ratio effect: (**a**) For various combination from 90:10 to 10:90, (**b**) the observation of the break-through location at skin/core ratio = 40:60.

**Figure 12 polymers-12-00024-f012:**
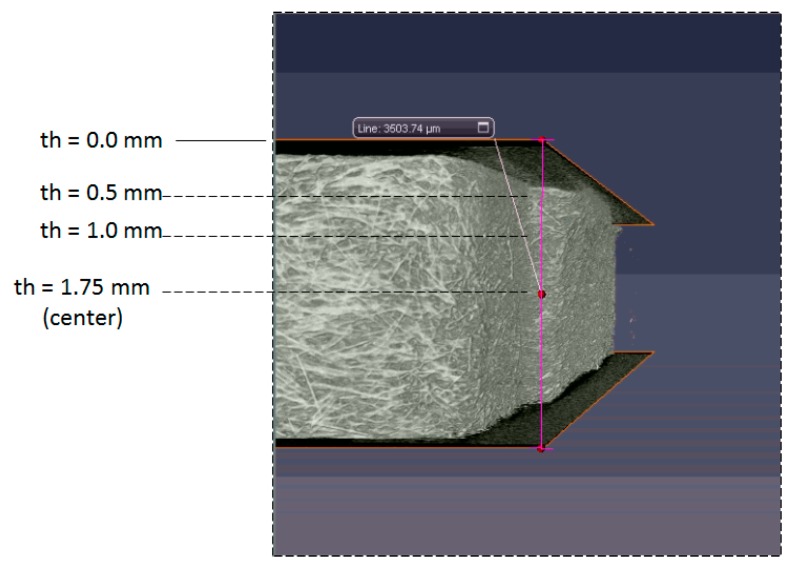
The locations for making the sliced plane to observe the fiber morphology; where th is the distance from the top surface.

**Figure 13 polymers-12-00024-f013:**
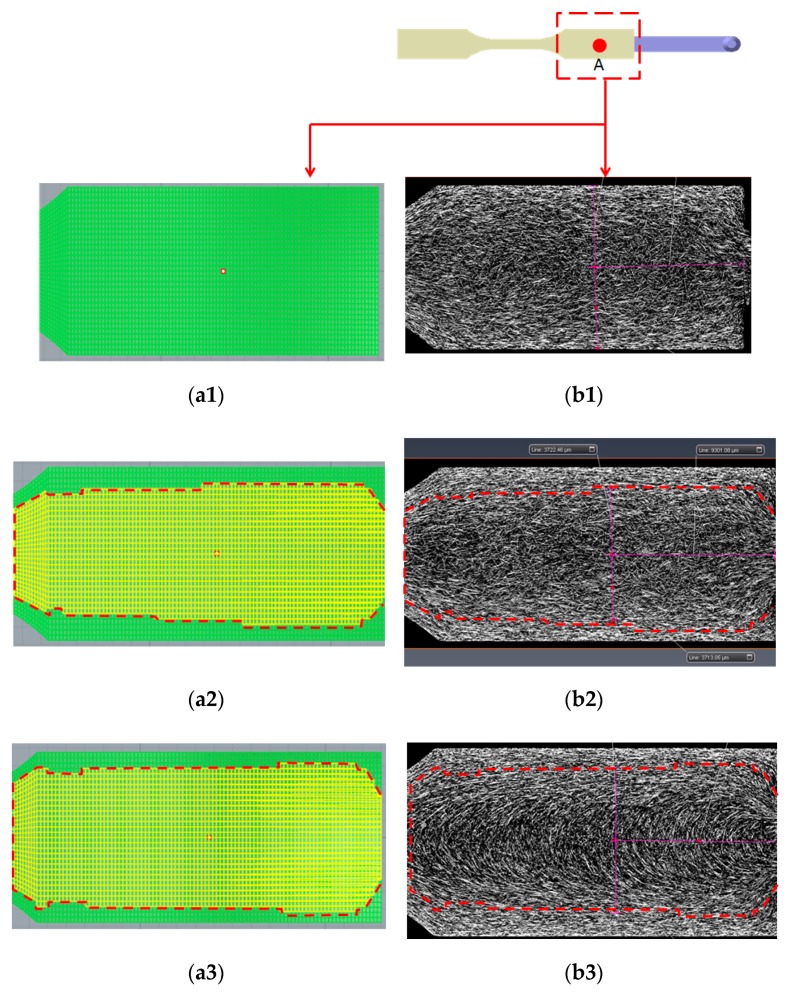
Observation of the fiber morphology based on the sliced plane for co-injected parts at near gate region (NGR) with different thickness locations: (**a1**) Simulation at th = 0.5 mm, (**b1**) sliced image at th = 0.5 mm, (**a2**) simulation at th = 1.0 mm, (**b2**) sliced image at th = 1.0 mm, (**a3**) simulation at th = 1.75 mm (central portion), (**b3**) sliced image at th = 1.75 mm.

**Figure 14 polymers-12-00024-f014:**
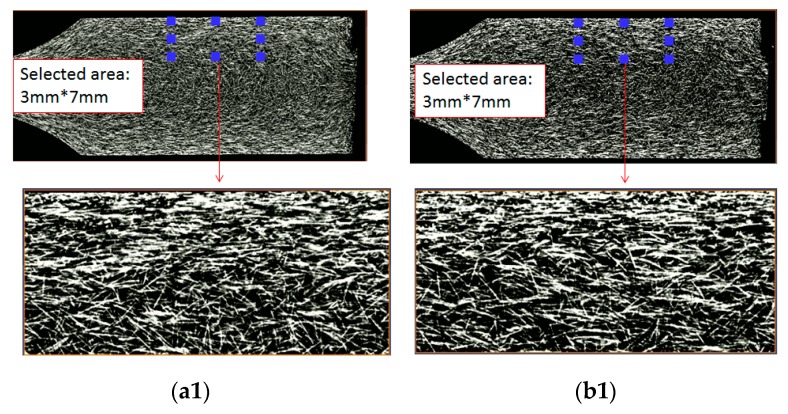
Observation of the fiber morphology based on the sliced plane for the single-shot and co-injected parts at point A with different thickness locations: (**a1**) Single-shot at th = 0.5 mm, (**b1**) co-injected at th = 0.5 mm, (**a2**) single-shot at th = 1.0 mm, (**b2**) co-injected at th = 1.0 mm, (**a3**) single-shot at th = 1.75 mm (central portion), (**b3**) co-injected at th = 1.75 mm.

**Figure 15 polymers-12-00024-f015:**
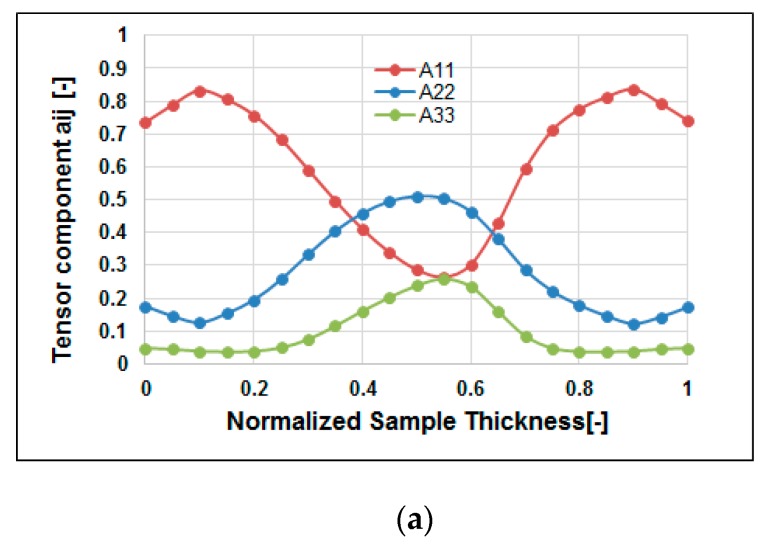
The comparison of the fiber orientation distribution (FOD) estimation: (**a**) Simulation at point A, (**b**) experiment around point A (NGR).

**Figure 16 polymers-12-00024-f016:**
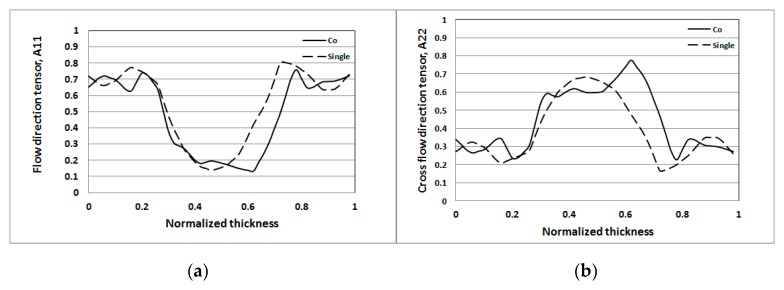
The comparison of the fiber orientation distribution (FOD) tensor component at NGR: (**a**) flow direction tensor A_11_, (**b**) cross-flow direction tensor A_22_.

**Figure 17 polymers-12-00024-f017:**
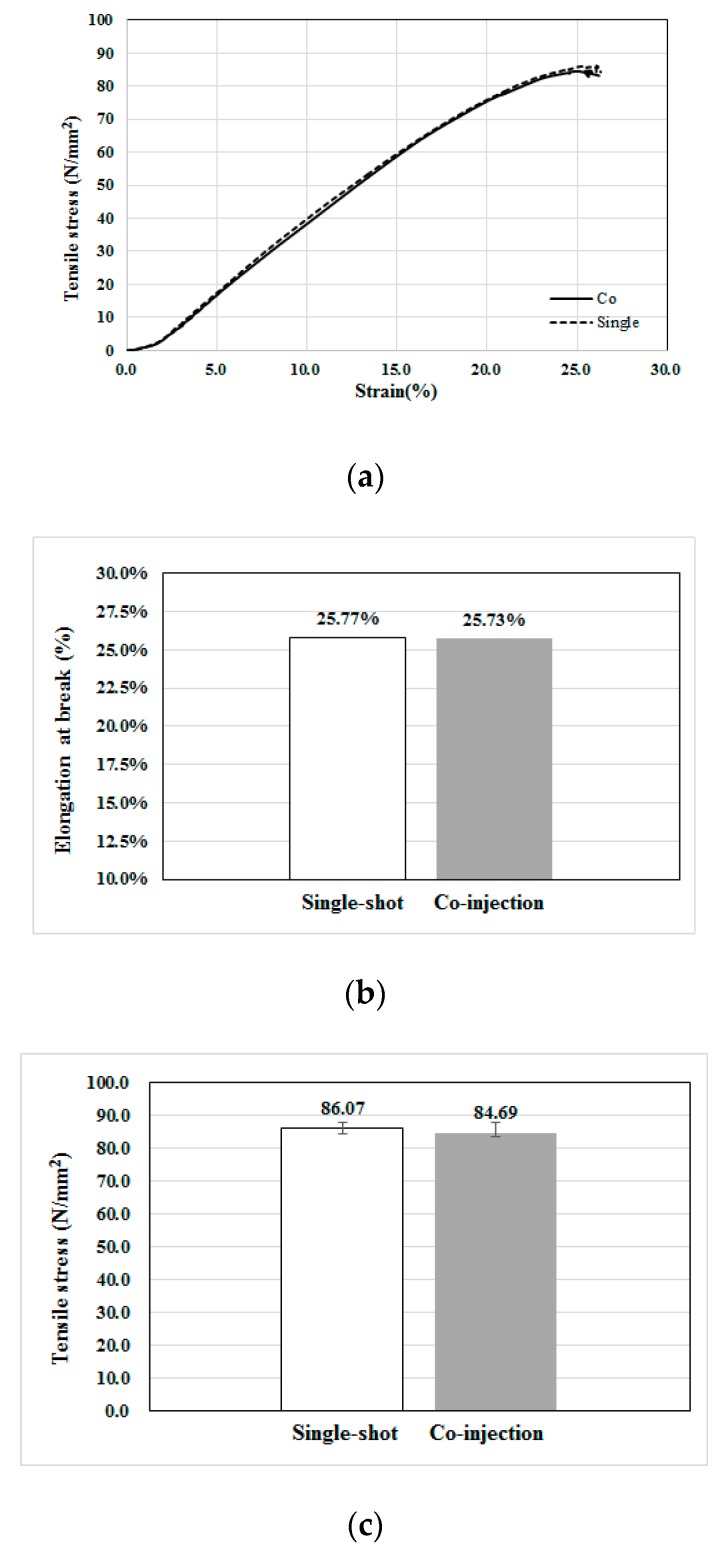
Tensile property measurement for single-shot and co-injection specimens: (**a**) The average tensile stress-strain behavior, (**b**) the average elongation at break, (**c**) the average tensile strength.

**Table 1 polymers-12-00024-t001:** The mesh types and related information.

	Mesh 1	Mesh 2	Mesh 3	Mesh 4	Mesh 5
mesh type (runner)	tetrahedron				
mesh type (cavity)	hexahedron				
layers in thickness	5	10	15	20	25
mesh size (mm)	0.2				
cavity count	51,520	103,040	154,560	206,080	257,600
runner count	22,940	40,560	49,340	46,640	66,900
total element count	74,460	143,600	203,900	252,720	324,500
analysis time (h)	0.25	0.45	0.61	0.75	1

**Table 2 polymers-12-00024-t002:** Process condition for co-injection.

Material	Skin: PP Globalene SF7351; Core: PP Globalene SF7351
Filling time (s)	0.3
Packing time (s)	----
Flow rate (cm^3^/s)	10
Melt temperature (°C)	230
Mold temperature (°C)	35
Injection pressure (MPa)	175
Core switch over (by volume filled) (%)	60

**Table 3 polymers-12-00024-t003:** Process condition for single shot injection.

Material	PP Globalene SF7351
Filling time (s)	0.3
Packing time (s)	3
Flow rate (cm^3^/s)	10
Melt temperature (°C)	230
Mold temperature (°C)	35
Injection pressure (MPa)	175

**Table 4 polymers-12-00024-t004:** Specimen Specification.

Symbol	Definition	mm
L_1_	Length of narrow portion	
L_2_	Distance between broad parallel portions	
L_3_	Initial clamping length	
L_4_	Overall length	
W_1_	Width of narrow portion	
W_2_	Width at ends	
H	Thickness	
R	Radius	

**Table 5 polymers-12-00024-t005:** The comparison of the tensile properties.

Item	Single-Shot	Co-Injection
Tensile Strength (N/mm^2^)	86.07	84.69
Tensile Modulus (N/mm^2^)	75.93	39.08
Elongation at Break (%)	25.77%	25.73%

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
