# Peer review of "Investigation on the Fiber Orientation Distributions and Their Influence on the Mechanical Property of the Co-Injection Molding Products"

_polymers, 2019, doi:10.3390/polym12010024_

Round 1

Reviewer 1 Report

The manuscript submitted studies the fiber orientation distributions and the fiber morphologies of single-shot and co-injection molding. The authors studied the fiber orientation distributions and their influence on the tensile strength for the single-shot and co-injection molding. The manuscript is within the scope of Polymers. The manuscript is generally well written. However, before publication some concerns must be addressed. 

a) There are too many long sentences in the abstract part, for the convenience of readers, it is suggested to replace them with several short sentences. b) The keywords selected in this paper can not well reflect the central content of the  manuscript. It is suggested that the author carefully consider the keywords again. c) The references cited in the introduction of the manuscript are far away from the present. In order to fully reflect the latest research status of researchers around the world, it is suggested that the number of references in recent years can be increased appropriately. d) Thesizeof '(a)' below the Figure 1 is different from others, Please correct it. e) According to the test data of tensile strength measurement in this manuscript, the expression of 349 lines 'the fiber orientation tensor at flow direction (A11) of the single-shot is much higher than thatof the co-injection'is not accurate, so it is suggested to remove the 'much'. f) In order to correspond with the writing order of the manuscriptand make the writing more fluent, it is suggested that the author adjust the placing order of the four conclusions in the conclusions part.

Author Response

Thank you very much for your priceless comments and suggestions. Regaring our answer, please see the attachment.

Reviewer 2 Report

Review of the paper entitled “Investigation on the Fiber Orientation Distributions and Their Influence on the Mechanical Property of the Co-injection Molding Products”

In this paper the fiber orientation tensors were investigated and compared in single and co-injection molded products. The paper contains convincing results, but the evaluation of the measurement results should be improved.

The following points should be dealt with before publication:

1, First of all, the authors should highlight the originality of this article. Please compare these results with previous studies and report their new insights.

2, Please, give some references to the equations.

3, Please give some details about the meshing of the simulation model (element size, meshing strategy, mesh type…). The mesh resolution and the element size could influence the results.

4, Please give the explanation of the orientation tensors. A figure can help to understand its meaning.

5, In line 209 the reference connects to figure 5 (c), not 5 (a).

6, In chapter 4.3.1. and in figure 6. please give the skin-core ratio, which was used for the validation process.

7, In my opinion in chapter 4.3.4. the values of orientation tensors are incorrect. Please revise them.

8, Near the top surface (from normalized thickness 0 to 0.2) of the specimens the trend of the orientation tensors is different in case of simulations and experiments. What is the reason of this difference?

9, Is the difference between the mechanical properties of single and co-injection molded samples significant? The difference is under 5%. Other mechanical properties should be determined, like tensile modulus, flexural strength and flexural modulus.

10, In general, the paper contains some grammatical errors and mistyping. Please use the same style for the figures.

Author Response

Thank you very much for your wonderul suggestions.  Regarding the answer for questions, please see the attachment.

Round 2

Reviewer 2 Report

The manuscript was resent after a full revisions. The answers are clear and detailed. I can accept the manuscript with this content.